# Techno-Economic Optimization Study of Interconnected Heat and Power Multi-Microgrids with a Novel Nature-Inspired Evolutionary Method

Paolo Fracas [1,2,*], Edwin Zondervan [2], Meik Franke [2], Kyle Camarda [3], Stanimir Valtchev [4] and Svilen Valtchev [5,6]

1    Genport srl–Spinoff del Politecnico di Milano, Via Lecco 61, 20871 Vimercate, Italy
2    Faculty of Science and Technology, University of Twente, Drienerlolaan 5, 7522 NB Enschede, The Netherlands
3    Department of Chemical and Petroleum Engineering, University of Kansas, 1530 West 15th Street, Lawrence, KS 66045, USA
4    CTS UNINOVA, University NOVA of Lisbon, Campus FCT, 2829-516 Caparica, Portugal
5    CEMAT-IST, University of Lisbon, Av. Rovisco Pais 1, 1049-001 Lisbon, Portugal
6    ESTG, Polytechnic of Leiria, Campus 2, Morro do Lena-Alto do Vieiro, 2411-901 Leiria, Portugal
*    Correspondence: paolo.fracas@genport.it

**Abstract:** The world is once again facing massive energy- and environmental challenges, caused by global warming. This time, the situation is complicated by the increase in energy demand after the pandemic years, and the dramatic lack of basic energy supply. The purely "green" energy is still not ready to substitute the fossil energy, but this year the fossil supplies are heavily questioned. Consequently, engineering must take flexible, adaptive, unexpected directions. For example, even the natural gas power plants are currently considered "green" by the European Union Taxonomy, joining the "green" hydrogen. Through a tight integration of highly intermittent renewable, or other distributed energy resources, the microgrid is the technology of choice to guarantee the expected impacts, making clean energy affordable. The focus of this work lies in the techno-economic optimization analysis of Combined Heat and Power (CHP) Multi-Micro Grids (MMG), a novel distribution system architecture comprising two interconnected hybrid microgrids. High computational resources are needed to investigate the CHP-MMG. To this aim, a novel nature-inspired two-layer optimization-simulation algorithm is discussed. The proposed algorithm is used to execute a techno-economic analysis and find the best settings while the energy balance is achieved at minimum operational costs and highest revenues. At a lower level, inside the algorithm, a Sequential Least Squares Programming (SLSQP) method ensures that the stochastic generation and consumption of energy deriving from CHP-MMG trial settings are balanced at each time-step. At the upper level, a novel multi-objective self-adaptive evolutionary algorithm is discussed. This upper level is searching for the best design, sizing, siting, and setting, which guarantees the highest internal rate of return (IRR) and the lowest Levelized Cost of Energy (LCOE). The Artificial Immune Evolutionary (AIE) algorithm imitates how the immune system fights harmful viruses that enter the body. The optimization method is used for sensitivity analysis of hydrogen costs in off-grid and on-grid highly perturbed contexts. It has been observed that the best CHP-MMG settings are those that promote a tight thermal and electrical energy balance between interconnected microgrids. The results demonstrate that such mechanism of energy swarm can keep the LCOE lower than 15 c€/kWh and IRR of over 55%.

**Keywords:** multi-microgrid; optimization; evolutionary algorithm; differential evolution; SLSQP

## 1. Introduction

A microgrid (MG) is a controllable, independent small energy system comprising distributed generators (DG), loads, energy storage (ES) systems, and control devices [1].

The combined cool, heat and electric power microgrid (CHP-MG) systems, otherwise called Three-Generation systems, can improve the quality of energy, generated from renewable power sources, such as wind turbines and solar panels. Compared with conventional

CHP systems, CHP-MG have greater functionality, because they do not only satisfy cooling-, heating- and electric power demands (e.g., for residential, commercial, industrial and university buildings), but they also interact with the main grid to provide energy demand response services, i.e., reserve, peak-shaving, load shifting, load shedding, and improved capability for the integration of non-dispatchable, i.e., renewable energy sources (RES) [2].

When two or more CHP-MG systems are interconnected into a multi-microgrid (CHP-MMG), they become flexible and capable to counterbalance the constantly varying conditions, thus matching the supply and demand at any time [3]. Any energy production surplus (i.e., exceeding the load demands and energy services) can be converted from electricity to heat and transferred to the nearby CHP-MG. Consequently, the total costs of ownership (TCO) for the installations are minimized, a better LCOE is achieved, and the IRR is higher than the obtained in a stand-alone MG [4].

Unpredictable and variable climatic conditions make the production of the energy with RES sporadic and not programmable, often even not predictable [5]. Matching the intermittent production of energy with the load profile uncertainty, grid (un-)availability, and (un-)flexible energy demand, requires new algorithms to solve these mostly stochastic optimization problems.

The CHP-MMG merged systems operation has an important, economic aim, together with their technical viability that must be proved. The techno-economic proof of the CHP-MMG requires optimization algorithms that will select the best among different solutions for the design, sizing, siting, and above all will select the best setting that maximizes the financial performance. Several works may be named that deal with the optimization of the operations of a single MG [6]. The Multi-microgrid optimization problems, however, are rarely investigated. Typical techno-economic analysis of a MG usually deals with the minimizations of the costs; stochastic models are approximated with linear models representing different finite scenarios. The simplified models are usually solved with mixed-integer linear programming (MILP) with a top-down approach (i.e., from the scenario to the solution).

In this work, it is proposed to solve the optimization problem without approximations in finite scenarios. The latter is a complex stochastic multi-objective problem that requires high computing resources. To overcome this problem, a novel two-layer optimization architecture is proposed to simultaneously find the optimal setting of the CHP-MMG. Moreover, a further developed method is implemented to validate the best probabilistic solution over all scenarios under realistic operating conditions.

At the inner layer, for a given trial setting, the hourly energy balance among unpredictable-intermittent production with renewable energy sources (RES), grid availability, flexible energy demand, uncertain load profile, is achieved with minimum operational costs and highest revenues with the Sequential Least Squares Programming (SLSQP) method. The trial setting is given by a novel evolutionary computing method that searches the best type (design optimization), size (sizing optimization) of distributed generators (DG), the best geo-location (siting optimization) to minimize LCOE and maximize IRR.

To conclude, the innovation contents of this work comprise: (1) a novel simulation-optimization tool designed to simultaneously determine under realistic operating conditions: the design, the sizing, the siting, the operation optimizations of two interconnected heat and power microgrids; (2) a two-layer optimization algorithm combining a novel evolutionary computing method with SLSQP to improve the quality of the results and the solution space search. The results of the simulations carried out in this paper have shown the beneficial effect induced by the collaboration scheme between the two interconnected heat and power microgrids, which enhances the overall efficiency, response to uncertainty and, consequently, better financial performance under the most likely scenarios. To the best of the writer's knowledge, interconnected heat and power multi-microgrids have not yet been thoroughly studied in the literature. Furthermore, techno-economic simultaneous optimization of siting, sizing, and operation of CHP-MMG has still not been addressed so

far in detail by other scientific studies. This constitutes another relevant contribution of this work.

*Features of Evolutionary Algorithms*

To find the optimal design, sizing, and siting of CHP-MMG, a stochastic multi-objective optimization (MOP) problem must be solved. This is a non-convex, non-linear problem that does not have a unique solution. Many solutions with incommensurable quality, hereafter called "probabilistic best solutions" (PBS) can be identified. Metaheuristics are useful for solving MOP [7]. The term "metaheuristic" [8] refers to a nature-inspired robust searching mechanism that may provide a sufficiently good solution. Evolutionary algorithms (EAs) are a family of random-based solution space search algorithms. An evolutionary algorithm is a special computation technique that draws inspiration from the principle of natural evolution and survival of the fittest that are described by Charles Darwin in his Theory of Evolution. That theory explains the principle of natural selection, which promotes the survival of species that are matched with their environmental conditions and explains the evolution of species as an outcome of random variations and natural selection. There are a wide variety of metaheuristics including genetic algorithms (GAs), evolution strategies (ES), evolutionary programming (EP), genetic programming (GP), and differential evolution (DE). Since EAs deal with a group of candidate solutions, it seems natural to use them in optimization problems to find a group of promising solutions. Indeed, EAs have proven very efficient in solving complex multi-objective optimization problems [9]. EA is a direction-based search method that optimizes a problem by iteratively trying to improve a candidate solution with respect to a given measure of quality. The metaheuristic EA method needs few or no assumptions about the problem and can search for solutions in very large spaces. Thus, EA are suitable when solving complex unconstrained global optimization problems. A population of target solutions evolves at each generation in new candidate solutions by combining existing ones and then keeping whichever candidate solution has the best fitness. The optimization problem is treated as a "black box" that merely provides a measure of quality given by the candidate solution, while the gradient is not needed. In 1995, R. Storn and K.Price [10] proposed the Differential Evolution (DE) algorithm, a stochastic population-based search method, for solving non-linear, high-dimensional, and complex computational optimization problems. DE uses a simple mutation operator based on differences between pairs of solutions (called hereafter vectors or individuals) to promote a search direction-based approach. It is worth noting that DE can support the contour matching, i.e., the vector population adapts such that promising regions of the objective function surface are investigated automatically once they are detected. To this end an important ingredient is the promotion of basin-to-basin transfer, where search points may move from one local minimum, to another [11]. DE also utilizes a mechanism, where the newly generated offspring competes only with its corresponding parent (previous vector) and replaces it if the offspring has a higher fitness value.

## 2. State of Art of Evolutionary Algorithms Applied to Microgrids

In the recent years, evolutionary algorithms (EA) have emerged as successful alternatives to more classical approaches for solving microgrid optimization problems. The applications of EA spun from optimization of management, scheduling of energy flows (operations), to optimal sizing of nominal power of generators and capacity of energy storage. Most research is dealing with specific sites of the energy systems. A few methods are considered and applied to determine optimal size and location of MG. Such methods can be categorized as single-objective and multi-objective sizing and siting optimization problems.

Single-objective methods target to minimize the cost or maximize the profits. Whereas for multi-objective methods are used to determine Pareto-optimal solutions balancing different objectives at the same time such as maximizing the power system reliability (i.e., minimize the energy underflows), minimizing the cost and the emission of pollutions, and maximizing the financial returns of investments (i.e., NPV, IRR).

### 2.1. Evolutionary Computing for Optimal Operations

Several works deal with the optimization of MG operations using different meta-heuristics. Such methods search for solutions over a well-defined MG configuration and installation site. These methods incorporate voltage, current and power flow regulations, multi-grid dispatch, pollutant emissions, reactive power, energy scheduling and the cost of energy. The work of H. Vahedi et al. [12] focused on developing a cost optimal operational strategy for a single MG using a differential evolution to meet the customer demand and ensuring system safety. M. Hemmati et al. [13] presented a comprehensive operational model for MGs in the islanded mode. A new learning-based differential evolution algorithm is presented to solve the operational problem of various DERs. W. Gu et al. [14] presented a review of the energy management of combined cooling, heating, and power (CHP-MG) microgrid with distributed cogeneration units and renewable energy sources which yield an effective solution to energy-related problems, including high energy demand, costs, supply security, and environmental concerns. J. Zhang et al. [15] proposed an optimal day-ahead scheduling model for a microgrid system based on a hybrid harmony search algorithm with differential evolution (HSDE). S. Reddy et al. [16] discussed a power scheduling approach for standalone MG. The scheduling problem is solved using a hybrid differential evolution and harmony search (HSDE) algorithm. M. Marzband et al. [17], developed an algorithm for an energy management system (EMS) based on multi-layer ant colony optimization approach (EMS-MACO) to schedule energy in MG. This algorithm can be used to determine the required load demand with minimum energy cost in a local energy market. N. Nikmehr et al. [18] applied a particle swarm optimization (PSO) to minimize the costs of MMG. The stochastic model of small-scale energy resources and load demand of each microgrid is developed to determine the best economic operation for each MG, based on the power transactions between the MG and main grids. The proposed methods allow to regulate the power demand and power transaction between each MG and the main grid. M. Hossain et al. [19] demonstrated an application of PSO for real-time energy management of a community microgrid. The complexity of time-varying electricity prices, stochastic energy sources and power demand is managed to save costs and minimize energy waste. Y. Li et al. [20] have investigated how to coordinate several scheduling objectives from the perspectives of cost, environment, and users, with a multi-objective dynamic dispatch model. An evolutionary algorithm is used to find a set of Pareto-optimal solutions. The results demonstrated the effectiveness of the suggested approach. Ferreira et al. [21] proposed a multi-objective optimization method structured in multi-layers to operate DER that need to be connected unevenly throughout the phases of a microgrid. An EA solves a multi-objective problem of maximizing the active power generation by single-phase distributed energy resources and minimizing the reactive power flow through the grid and grid currents unbalance at the point of common coupling. The results demonstrated the effectivity of the proposed approach to steer grid power flow and prioritize active power injection or compensation of currents unbalance.

More recently, H. Karimi et al. [22] proposed an analytic approach to identify the best coalition among microgrids in multi-microgrid systems. The proposed model evaluates all the strategies and ranks them from the viewpoint of MG. The algorithm enjoys a cooperative game and provides a stable coalition for microgrids. In this cooperative game, the overall gain of each strategy is divided among the cooperator microgrids based on their contributions. The results of case studies demonstrate that the proposed coalition formation significantly reduces the operating cost of the system and the amount of involuntary load shedding.

H. Peng et al. [23] proposed a new micro multi-strategy multi-objective ABC algorithm to solve multi-objective evolutionary algorithm for microgrid energy optimization, called μMMABC. The latter is used to divide the population into multiple subgroups and produce offspring in parallel to balance the exploration and exploitation. In addition, an adaptive updating mechanism is proposed to renew the population adaptively. The mechanism can adaptively select more convergent and diverse solutions at different stages to balance the

exploration and exploitation of the algorithm. Furthermore, to improve the performance of µMMABC on problems with irregular Pareto-fronts, the reference point reconstruction with intermediate strategy is also proposed. The experimental results show that the proposed algorithm is more competitive and effective than the traditional multi-objective evolutionary algorithm.

Q. Wu et al. [24], proposed a multi-stakeholder benefit optimization method based on Stackelberg game theory, and establish a local market energy transaction model. The model is composed of external distribution network, leader-intermediary agent (IA) and follower-microgrids (MGs) and uses the master–slave game method to solve the optimization strategy. Under this mode, the profit of each stakeholder in MMG increased by 2.64%, 4.24%, 1.38% respectively. The case results show that based on the method in this paper, multi-stakeholder in MMG can improve economic performance and can increase the level of energy autonomy of MMG.

### 2.2. Evolutionary Computing for Optimal Siting and Sizing

Similar to the optimization of operation problems, in the last ten years, metaheuristics have been proposed to solve multi-objective siting of MG. Notably, multi-microgrid optimization problems are rarely investigated. Sizing problems are usually combined with a low-level management of operations, which are usually solved as MILP. H. Doagou-Mojarrad et al. [25] introduced an interactive fuzzy method, to solve the problem of optimal placement and sizing of DGs in a distribution network. The multi-objective function based on electrical energy losses, cost and pollutant emissions is handled by an evolutionary algorithm. B. Li et al. [26] proposed a combined sizing and energy management methodology, formulated as a leader-follower problem. The leader problem focuses on sizing and selects the optimal size for the microgrid components. The energy management issue is translated into a unit commitment problem and is solved as a mixed integer linear program. Uncertainties are considered using a robust optimization method. Several scenarios are modeled and compared via simulations to show the effectiveness of the proposed method. S. Mohseni et al. [27], investigated a novel multiagent-based method applied to the sizing of the components of an is-landed combined heating and power residential microgrid that includes a hydrogen refilling demand of the fuel cell electric vehicles (FCEV). The proposed architecture consists of five agents, namely a generation agent, an electrical- and thermal-loads agent, a FCEV refilling station agent, a control agent, and a design agent. The design is the main agent that according to its interactions with the control agent and by minimizing the total costs of the system through PSO, finds the optimal sizes of the system's components. In 2017, A. Kaabeche et al. [28] proposed a hybrid RES sizing method, taking into account the combination between the RES, the ES capacity and a given load profile. This optimization method is based on the Firefly Algorithm (FA), considering a load dissatisfaction rate (LDR) criterion, the electricity cost indicator for power reliability and system cost. J. Jung et al. [29] developed a technique for the planning and design of hybrid renewable energy systems in MG. A Distributed Energy Resources Customer Adoption Model (DER-CAM) determines the optimal size, type of DERs and their operating schedules. The electrical grid of the Brookhaven National Laboratory campus is used to demonstrate the effectiveness of this approach. A.M. Ramli et al. [30] optimized the size of hybrid microgrid system components, including storage, to determine system cost and reliability. The optimal sizing of a PV/wind/diesel HMS with battery storage is conducted using a Multi-Objective Self-Adaptive Differential Evolution algorithm. The objectives are treated simultaneously and independently, thereby leading to a reduction in computational time. Results show that a set of design solutions could assist researchers in selecting the optimal MG configuration. N. Ghorbani et al. [31] presented a hybrid genetic algorithm based on PSO applied to the optimal sizing of an off-grid house with photovoltaic panels, wind turbines, and battery. The minimization of the total costs of ownership was the main goal of this study. D.R. Prathapaneni et al. [32] proposed a leader-follower based design optimization method for microgrids where the management of load demand is incorpo-

rated into the sizing process. A microgrid powering a desalination plant is considered to evaluate the performance of the proposed method. The results demonstrated that the proposed coordinated sizing is cost-effective, and it provides better operational flexibility.

More recently, I. Cetinbas et al. [33] present a new hybrid metaheuristic algorithm, the hybrid Harris Hawks Optimizer-Arithmetic Optimization Algorithm. It is proposed for sizing optimization and design of autonomous microgrids. The proposed hybrid algorithm has been developed based on operating the Harris Hawks Optimizer and the Arithmetic Optimization Algorithm in a uniquely cooperative manner. The developed algorithm is expected to increase the solution accuracy by increasing the solution diversity during an optimization process. The hybrid algorithm is tested on a microgrid that consists of a photovoltaic (PV) system, a wind turbine (WT) system, a battery energy storage system (BESS), diesel generators (DG), and a commercial type of load. For the optimal capacity planning of these components, a problem in which the loss of power supply probability and the cost of energy, are defined as the objective function, is formulated. The optimization undertaken by the proposed algorithm has produced the lowest loss of power supply probability and lowest cost of energy along with the highest rate of renewable fraction.

B.O. Alawode et al. [34] presented a flexible method for the optimal sizing and operation of Battery Energy Storage System (BESS) in a wind-penetrated microgrid system using the butterfly optimization (BO) algorithm. The butterfly optimization algorithm was utilized for its simple and fast implementation and for its ability to obtain global optimization parameters. In the formulation of the optimization problem, the study considers the depth of discharge and life cycle of the BESS. Simulation results for three different scenarios were studied, analyzed, and compared. The resulting optimized BESS connected scenario yielded the most cost-effective strategy among all scenarios considered.

In a further study, E. Naderi et al. [35] proposed an optimal energy management and sizing of the microgrid based on Mixed Integer Linear Programming (MILP). The Monte Carlo method is employed to model and estimate wind behavior. Also, for regulating production and demand in the microgrid the demand response program is conducted to improve the contribution of the renewable energy resources. The planning is constructed as an optimization problem. By solving it, the size and production magnitude of energy sources, as well as storage conditions, are determined. Finally, the proposed method is simulated for all seasons of two scenarios. The results confirm the desired energy management and cost reduction in the studied grid.

In this work, the techno-economic optimization study is extended to multiple interconnected heat and power microgrids. Specifically, the study focuses on finding the optimal location, sizing, and operation. The goal is to achieve the highest IRR and lowest LCOE, depending on energy policies, cost, and hydrogen type in on-grid and off-grid contexts.

### 3. The Two-Layer Optimization Method

A two-layer algorithm to simultaneously find the optimal design, site, sizing, and operation of CHP-MMG is proposed. The inner layer is a convex piecewise-linear problem and is solved with the SLSQP method. The outer layer simultaneously solves a nonlinear, non-convex problem, with two novel evolutionary methods (i.e., ADE, AIE). The fitness values are generated by an analytical techno economic (ATE) model. A detailed description of ATE is reported in the publication of the authors P. Fracas et al. [36]. Figure 1 shows the flowchart of the algorithm. ATE incorporates a set of DER models, the SLSQP algorithm, the associated loss objective function, energy balances and boundaries dictated by outer ADE/AIE methods and DER's state of health. The SLSQP method ensures that the generation and consumption of energy are balanced each time-step $t_s$ (i.e., hour) at minimal operational costs and highest revenues streams. The objective function of the dispatch problem to solve with SLSQP can be defined as:

$$\min f(x) = \sum_{c=1}^{N} LCOE_c \cdot x_c(t_s) - \sum_{r=1}^{R} LSOE_r \cdot x_r(t_s) \tag{1}$$

where $x_c$, $x_r$ are the elements of the X-array candidate solution (i.e., respectively the energy flows of DERs consuming and generating energy). The LCOE is the ratio between the TCO, and the energy generated along the lifetime of the DER. Similarly, the LSOE is the selling price deducted from the LCOE of the microgrid's products. While minimizing at each time-step the cost-revenue streams, the electric and the thermal generation of microgrids and interconnections should balance the demands surplus DER and networks losses. To avoid energy under-flows and improve the computing convergence, the constraints are inequalities both for the thermal and electric energy flows of the DER and equalities for the exchanged energy flows between the two MG:

$$\begin{cases} \left( \sum_{i=1}^{N} g_i^{der}(t_s) \right) \cdot k_d - \left( \sum_{j=1}^{M} g_j^{load}(t_s) \right) \cdot k_l \geq 0 \\ \sum_{h=1}^{2} g_h^{itc}(t_s) = 0 \end{cases} \tag{2}$$

where the terms $g^{der}$ and $g^{load}$ are the energy flows of the DER and $g^{itc}$ are the energy flows between the MGs. The index $i$, $j$ and $h$ specify respectively each generator, load, and device to interconnect the MG. The $k_d$, $k_l$ are parameters to calibrate the contribution of the two terms and hence, to set the size of the overflow. The latter permits the algorithm to improve the capability to converge. The DER's models dynamically adapt the boundaries and computes the states of DERs after the elaboration of SLSQP. At the end of the iterations (e.g., 8760 time-steps which are equivalent to one year), ATE exits the SLSQP loop and computes the actual key financial criteria (LCOE, IRR, NPV) over the lifetime of the installation.

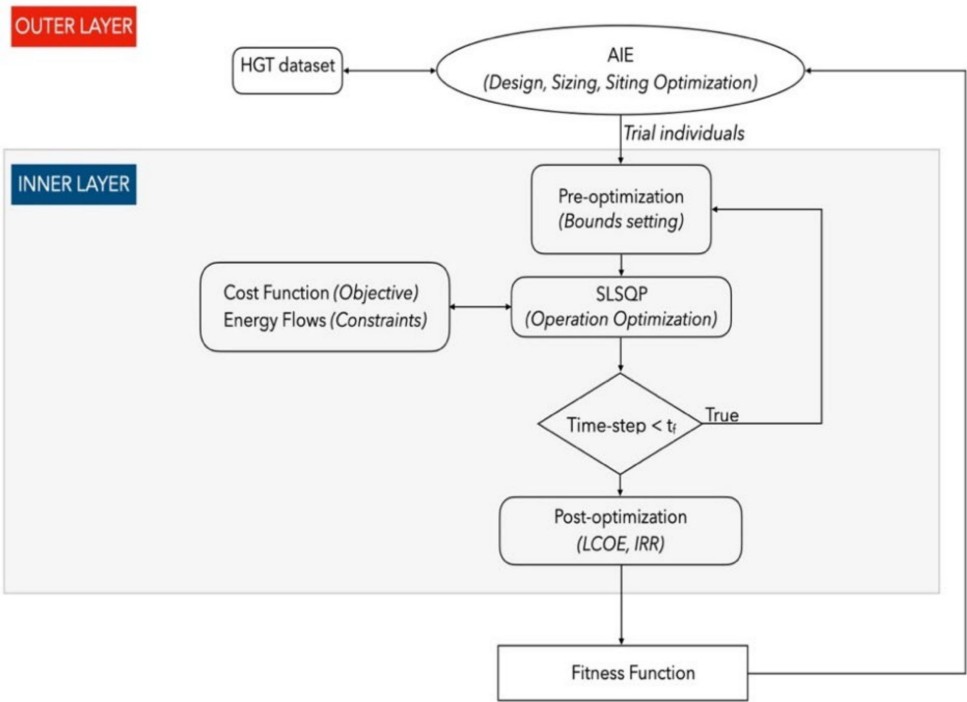

**Figure 1.** Framework of the two-layer optimization method.

The evolutionary algorithm at the top, generates the trial setting comprising the size of DER's size, the latitude, and the longitude. The setting is used by the ATE model for setting the boundaries of SLSQP method.

### 3.1. Introduction to Self-Adaptive Evolutionary Algorithms

Two novel self-adaptive evolution algorithms, hereafter named ADE and AIE have been developed and integrated to optimize the search of the best CHP-MMG setting. Before proceeding to discuss the overall structure of the algorithm, the first step is to define the objects, i.e., the candidate solutions within the search area.

An individual is a term used to denote an object. The individual is composed by chromosomes; each of them embodies several elements named genes. The genotype is the set of genes. The phenotypes are the set of characteristics of an individual, as they result from the expression of its genotype. In our problem, the phenotype expresses the observable behavior of individual interacting with local stochastic environmental conditions (e.g., radiation, wind speed, load profile, grid outages). In our problem, the individual is composed by 38 genes. The genotype contains two chromosomes: the first (chromosome of siting) embeds two genes describing the geo-spatial location (i.e., latitude, longitude) of the CHP-MMG configuration; the second (chromosome of sizing features) is composed by 36 genes that are describing the size of each DER. The two chromosomes (genotype) are the first property of the individual; the second property is the measure of its quality (fitness value). The individual can also be defined as a list of data having one dimension.

In our evolutionary computing problem, a population embedding 12 individuals is considered. The initial population is randomly generated according to a uniform distribution. After initialization, EA enter a loop of evolutionary operations: adaptive mutation, crossover, and selection. At each generation, a mutant genotype is created with three individuals, named parents, randomly selected in the mating pools. The target vector is perturbed with the differences of the parents. The resulting difference vector is scaled down with a mutant factor which is self-adapted to the difference vector based on the latest best-so-far fitness values. This approach enables optimization of the search radius and increases the convergence of the algorithm. In AIE the mutant vector is obtained with a normal random distribution whose standard deviation and mean are driven by the diversity of population and fitness convergence. Chromosomes of the mutant are randomly recombined with the target individual and with individuals randomly chosen in a dataset (horizontal gene transfer). In ADE, the genes of the mutant individual are mixed and recombined with the target individual to obtain a trial individual. The intensity of crossover is adapted to the difference vector based on the best-so-far fitness values.

In both algorithms, a one-to-one survivor selection criterion is used to find the best individual. The selection criterion is based on the fitness values. The trial individual competes with the target vector. The individual with the lowest fitness value survives into the next generation. The selection procedure selects the better one between the target vector and the trial vector. Individuals with higher quality have a higher probability of being selected into the mating pool so that the good ones will have more chances to breed. Then the newly generated population replaces the old one and another generation starts. In the development of the ADE and the AIE algorithms, the above-mentioned techniques keep the genotype diversity within an optimal bandwidth. The implementation is based on adapting the mutant, crossover parameters and a switching strategy to generate the mutant vectors and to transfer the genetic code during the recombination phase.

### 3.2. Self-Adaptive Differential Evolution Algorithm

The self-adaptive differential evolution algorithm (ADE) proposed method derives from DE. Compared to the original version introduced by Storn et al. [10] here, the mutant and crossover parameters are adapted according to the difference of the most recent best-so-far fitness values. A detailed description of ADE is reported in the publication of the author P. Fracas [36]. The difference vector distribution usually adapts to the landscape of the objective function. During the trials, it has been observed that a single difference vector limits the potential perturbation possibilities for a base vector and hence stagnation in a certain space search area may occur, which leads away from the global optimum.

Perturbation of the base vector by mutation has been treated very early and has led to various variants of DE. To enhance genotype diversity, the original mutation strategy is modified, by perturbing the base vector $X_{1,g}$ with three difference vectors whose intensity is adapted. The goal is to generate more distributed difference points without increasing the number of population members. The variant of the original mutant equation is proposed in the following Equation (3):

$$V_{i,g} = a_{i,g}^{par} \cdot X_{1,g} + a_{i,g}^{dith} \cdot \frac{(X_{2,g} - X_{3,g})}{3} + a_{i,g}^{dith} \cdot \frac{(X_{1,g} - X_{2,g})}{3} + a_{i,g}^{dith} \cdot \frac{(X_{1,g} - X_{3,g})}{3} \quad (3)$$

where $a_{i,g}^{par}$, $a_{i,g}^{dith}$ are the self-adaptive parameters.

To set $a_{i,g}^{par}$, $a_{i,g}^{dith}$, a 1d-array is created with the subset of the latest best-so-far fitness values as follow:

$$F_g = [f_k^{bsf}, \ldots, f_g^{bsf}] \quad (4)$$

where $k$ is the index denoting a subset of $g$. Then, the corresponding 1d-array with the differences is calculated as follow:

$$S_{i,g} = [(f_k^{bsf} - f_{k-1}^{bsf}), \ldots, (f_g^{bsf} - f_{g-1}^{bsf})] \quad (5)$$

The difference vector ($S_{i,g}$) is used to self-adapt the parameters $a_{i,g}^{par}$, $a_{i,g}^{dith}$ and thus, to maintain an optimal population diversity.

$$a_{i,g}^{dith}, a_{i,g}^{par} c_{i,g} = \begin{cases} rand(v_{min}, v_{max}), & S_{i,g} \in S_{i,g} > 0 \\ rand(w_{min}, w_{max}), & S_{i,g} \in S_{i,g} \le 0 \end{cases} \quad (6)$$

Equation (4) to Equation (6) show that the mutant ($V_{i,g}$) and crossover parameters ($c_{i,g}$) are evenly distributed over intervals subject to the intensity and direction of the elements of $S_{i,g}$. Notably, $v$ and $w$, have been empirically set within an interval: [0.54, 0.97]. It has been observed that queries based on this bandwidth allow to obtain a suitable genotype diversity. The mutation vector $V_{i,g}$ is then mixed with the so-called target vector $X_{i,g}$ (where $i \ne 1, 2, 3$) through the classic variant of diversity enhancement, the crossover, which allow to properly mix the parameters of the mutation vector $V_{j,g}$ to generate the trial vector $U_{i,g}$.

The result of the objective function that yields a lowest result is the criteria to select which vector will survive between the trial and target vectors. The latter will become a member of the next generation g + 1. For each generation, the individual within the $g$-population having the best fitness i.e., the best-so-far (BSF) value, is preserved to keep track of the progress that is made during the minimization process.

### 3.3. Self-Adaptive Artificial Immune (AIE) Algorithm

In 2020 our life has been deeply influenced by the corona virus disease (COVID-19). The spread of this pandemic across the globe was an inspiration to imitate how immune system fights harmful virus that enter the body. The basic function for the immune system is to identify and destroy the virus. The pathogen is recognized by special structure molecules, known as antigen. An early example of autoimmune algorithm has been proposed by L. De Castro et al. [37]. Their algorithm mimics the clonal principle. In artificial immune algorithms the term of "individuals" is replaced by "antibodies", the objective function is replaced by "antigen", the fitness values with the term "affinity measure", the population with the "repertoire" and the mutation with somatic "hypermutation".

An innovative version of the EA, hereafter named self-adaptive artificial immune evolutionary algorithm (AIE) is proposed. This section provides a short description of this novel method and the results of a comparison analysis with a traditional Differential Evolutionary method. A detailed description of AIE is reported in the publication of the author P. Fracas [38].

In the early generations a mutant antibody is created with a random generator. In the later number of generations, a difference vector-based mutant (i.e., ADE) is randomly recombined with external antibodies. The latter is randomly chosen by the horizontal gene transfer (HGT) and vertical gene transfer (VGT) techniques. The first method to create a mutant object mimics the early-stage mechanisms to generate an antibody to unknown antigens. The subsequent method is based on the difference between a couple of antibodies. It imitates a more sophisticated mechanism to evolve an antibody in the acquired immune system. The recombination method (HGT/VGT) imitates the improvement of the immunity deriving from external genetic pieces (i.e., RNA) having beneficial properties. The parameters i.e., popsize, generations, boundaries are assigned to the matrix repertoire. Then, the main loop is repeated until the stop criterion (e.g., maximum generation, minimum affinity measure) is reached. HGT has been recently recognized to play a relevant role in the evolution of living organisms in combination to vertical transfers of genes. In biological eukaryotic and prokaryotic organisms share a large portion of non-coding DNA/RNA transferred with lateral mechanisms (e.g., virus). Parts of DNA/RNA from a dead, disintegrated organism can, in some rare cases, penetrate the cellular organism wall and be incorporated into living cell DNA. W. Rafajłowicz. in 2018 [39] proposed HGT to transfer genes from a random individual in old population. Numerical results indicate the usefulness of HGT when applied to optimization problems of moderate size. When the vertical/horizontal recombination is executed, the affinity measure is computed and hence, the loop ends. Hence, the *j-trial antibody* and the *j-target antibody* are evaluated and the antibody returning the minimum affinity measure is selected. Figure 2 shows the comparison between AIE, and the original DE algorithm proposed by Storn [11]. The results for AIE are represented with blue bandwidth and for DE with the red bandwidth. The vertical axis indicates the best-so-far fitness values found by the algorithm for each evaluation in each query. The horizontal axis indicates the number of fitness evaluations (NOFE). The curves aggregate the measurements at each iteration for the mean and the 95% confidence interval around the mean. The behavior of the curves demonstrates an overall better quality of solutions with AIE. The gap between the means values of the AIE (dashed blue line) and DE (dashed red line) widens since 120 generations. The overall results demonstrate the effectiveness of the strategy implemented in AIE. At the beginning, the random mutant enhances the initialization phase, then the mutant based on ADE combined with the HGT/VGT crossover enhances the search direction.

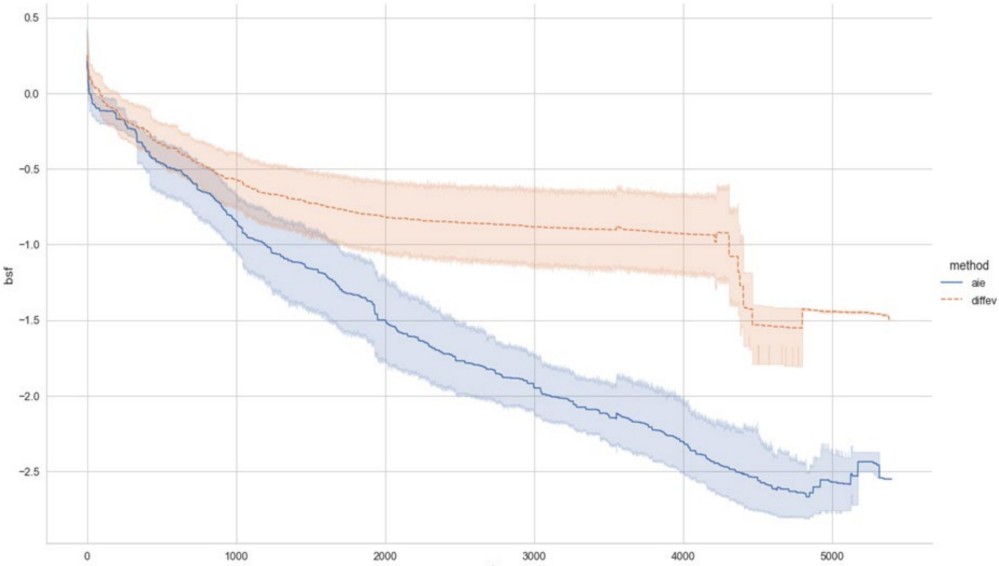

**Figure 2.** Comparison of fitness spread in 20 runs, 350 generation, ATE and AIE algorithms with analytical objective function.

The AIE method avoids that parents suffer from a loss of diversity, and thus it prevents that populations are stalled in a low-fitness valley. Diversity enhances crossover and enables crossover to be more effective than mutation. Diversity can be crucial in dynamic optimization, when the problem landscape changes over time [40]. After the initial generations, the search area is in a better position where it proceeds with the differential mutant vectors. The initial control of mutant and crossover, enhances the exploration of wide regions of the search space. Therefore, the risk that a repertoire converges a local optimum is mitigated. HGT gives a contribution to accelerate the search and it attenuates the fitness error. The method mimics the antibodies that randomly mutate and casually recombine through HGT with external organisms. This approach strengthens the ability to confer a protection against an aggressive antigen that evolves over the RNA. The mutation of the antigens is represented by the stochastic terms into ATE model that is part of the fitness function (i.e., Equation (7)). The aleatory recombination with pieces of external RNA, secretes quicker and better repertoire of antibodies. These mechanisms enable to successfully evolve generation by generation, to improve faster the affinity measures and to effectively converge into global minimum locations.

## 4. The Fitness Function

Each solution (i.e., the *j*-individual) of the *g*-population has associated a performance indicator: the fitness value. The latter is derived from a fitness function (FF) that is designed to search for the most profitable solution among a mix of DER type (design objective), size (sizing objective) and geo-locations (siting objective). FF is a multi- objective relation that combines three terms: the levelized cost of energy, the internal rate of return, and penalties on unusual configurations and energy unbalances. The IRR is used to rank the profitability of the potential investment in the j-solution. The latter measures the rate of return on an investment, calculated from the discount rate that equates the present value of future cash inflows to the CHP-MMG's total cost of ownership.

The LCOE gives the cost of generating electric and thermal energy for the CHP-MMG including the cost of the energy-generating system, including all the costs over its lifetime, initial investment (CAPEX), operations, and maintenance, cost of fuel (OPEX) [36]. The LCOE is a measure of the marginal cost (the cost of producing one extra unit) of electricity and heating over an extended period.

The combination of these two objectives and the penalties associated to undesired configurations is given in the following relation:

$$f\left(x_{j,i,g}\right) = k_a \cdot \sum_{j=1}^{m} P_j + k_b \cdot \sum_{i=1}^{n} LCOE_i - k_c \cdot \sum_{i=1}^{n} IRR_i \tag{7}$$

The term $P_j$ represents the penalties. The latter is computed by a logic function that returns the value: "one" if the trial settings are unusual (e.g., FC without hydrogen tank or geo-location in the sea). The terms $LCOE_i$ and $IRR_i$ are inferred from the processing of the underlying SLSQP-based optimization of the test setting operations.

The coefficients $k_a, k_b, k_c$ are different weights, while *i* represents the *i*-CHP-MMG. The latter are empiric, and their role is to properly balance the contribution of each term (i.e., LCOE, IRR, P). The term $k_a$ is set to a high value (e.g., $10^{26}$) if the trial setting is unusual, otherwise its value is set to: "zero". The latter approach proved to be the most effective in discarding unusual configurations with a minimum number of iterations. The other two coefficients $k_b, k_c$ are set to: "one" to equally offset the contribution of the IRR and LCOE indicators.

LCOE is widely used to compare the economic competitiveness of the energy mix [41]. This term is easy to understand and straightforward to apply, which makes it preferable for many energy policymakers. However, the method is not exhaustive from the energy business point of view. The LCOE approach does not consider revenue. Additionally, the LCOE does not consider equity and loan investors, which influence the energy sources'

economic attractiveness. LCOE and IRR are not in conflict, but they are not necessarily proportionally correlated. LCOE is correlated to the total cost of ownership per energy generated, while IRR is subjected to the contribution of each product to the revenue stream.

The Equation (7) measures both the financial objectives (i.e., LCOE and IRR) and penalties (P) associated to undesired configurations, underflows energies (i.e., unbalanced energy flows). The penalties, discourage the selection of individuals with unusual configurations (i.e., generators without tanks and vice-versa) and thus contributes to accelerate the search of the best LCOE and IRR. The weighted sum method gives a decision maker the possibility to assign the importance for each objective with the selection of $k_a, k_b, k_c$.

The correlation between the standard deviation of fitness array and population's genes reflects the quality of the sought design, and this correlation can be used as stop criterion of AIE/ADE iterations. Moreover, the FF incorporates stochastic terms (i.e., climate variables, grid outage, load profiles) [36]. Hence, this problem returns clusters of global solutions (i.e., PBS) having a similar fitness value. To find the best configuration, the optimization must be repeated until the standard deviation of the PBS's genotypes reaches the desired value. The PBS having the lowest fitness value can be assumed as the candidate of being the "best of the best probabilistic solution".

The final task is to verify with the sample average approximation method [42] if the latter solution returns the best expected performance over all the uncertain scenarios. This approach can be defined as "simulation-optimization" [7–43].

## 5. Results and Discussion

The first part of this work is addressed to investigate how the geographical location affects the size and performance of CHP-MMG configurations in on-grid and off-grid scenarios. Moreover, the study investigates how the cost and type of the hydrogen impacts the selection of the CHP-MMG settings.

Figure 3 depicts the overall scheme of the CHP-MMG. The labeled boxes represent the techno-economic models of the DER. The red lines indicate the thermal network while the blue lines represent the electric network. The two CHP-MMG exchange electrical energy through the ITCEL device and the thermal energy through the ITCH device. Each MG independently exchanges electrical energy to the main grid (GRID). The generation of energy from renewables is represented with the labels: PV (photovoltaic panels), WT (Wind turbines) and ST (Solar thermal collectors). Fuel cells (FC) and OG (traditional gensets) are the further options for distributed generations.

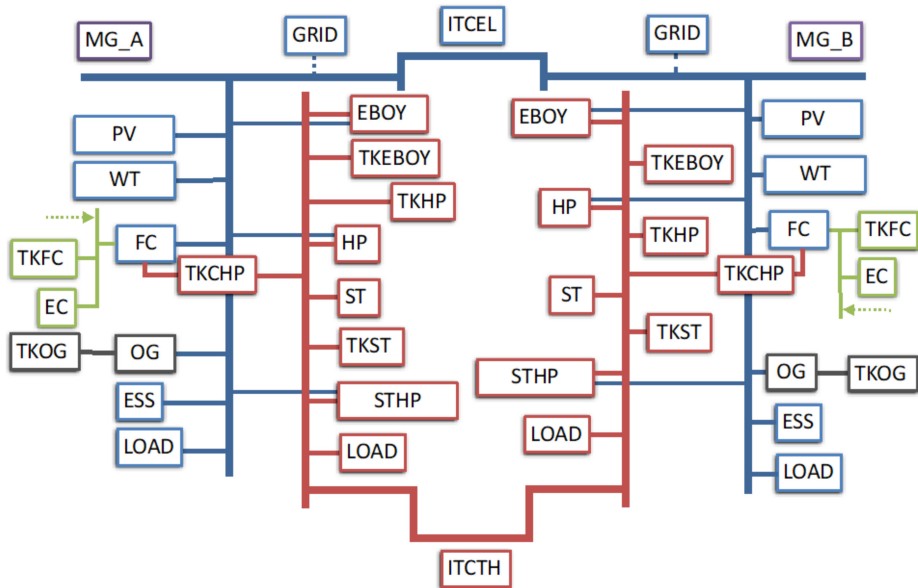

**Figure 3.** The overall architecture of two CHP-MMG comprising 36 DER, 2 electrical, thermal loads.

The electrical energy storage of energy is represented with the label ESS. Moreover, several thermal storages are indicated here with the initials 'TK' (i.e., tanks) followed by the name of the correspondent thermal generator. For example, the storage of hydrogen is indicated with the label 'TKFC'. The conversion from electrical to thermal energy occurs with two different types of thermal distributed generators: the electric boilers (EBOY) and the heat pumps (HP, STHP). The conversion for electricity to hydrogen occurs with the electrolyzers that are labeled: 'EC' while the DG labeled 'FC' are the fuel cells, converting the hydrogen into electricity and heat.

The overall techno-economic model embeds 38 variables from two different categories. These are the CHP-MMG settings to be optimized. The first group of 36 variables depicts the dimensions of the DER (i.e., 18 DER each CHP-MG) while the remaining 2 variables identify the geographic location in terms of latitude and longitude (i.e., the siting). The best setting must return the minimum actual levelized cost of energy (LCOE) and maximum internal return rate of investment (IRR). The SLSQP algorithm uses the techno-economic models to minimize the cost-profit objective function by computing at each time-step the optimal generation, storage, conversions, and exchange of thermal and electro-chemical energy, subjected to their size, state of health, nominal LCOE, LSOE and available energy from the renewable sources grid. It finally returns for each trial setting the input for the computation of the actual financial terms (i.e., LCOE, IRR). The optimal setting for the 38 variables is obtained as a solution of a highly non-linear and non-convex problem requiring stochastic optimization techniques. To fulfil this task, the novel ADE/AIE evolutionary optimization algorithm described in the Section 3 has been used.

This section provides the results of the study, carried on investigating how the scheme of collaboration between the microgrids based on an optimal combination between conversions and exchanges of thermal, electrical, and chemical energies to obtain the best financial performances.

The overall results demonstrate the benefit of interconnecting MG. Crossflows of thermal and electrical energies allow to convert and share fractions of the exceeding energy between the MG and thus DR may be downsized. For example, abundant electricity not used to power loads or charge batteries in a certain time interval is transferred to the other MG to be converted into heat energy. Then, some of the heat returns to the other microgrid.

Three options for supply of hydrogen are considered. The first is the on-site production by electrolyzers embedded into the CHP-MMG. The second is hydrogen sourced from large renewable-to-hydrogen (R2H2) plants. The third deals with on-site production by plasma assisted decomposition of methane.

R2H2 links WT and PV to electrolyzer stacks, which pass the excess electricity through water to split it into hydrogen and oxygen. In our simulations, hydrogen delivered by R2H2 plants is stored at the CHP-MMG site for later use with FC. R2H2 is the most promising scenario for generation of green hydrogen over the coming decade. For regions where renewable energy costs are on average higher, e.g., Northern Europe, there are areas with favorable conditions for larger-scale production of renewable energy. This makes it possible to produce hydrogen at lower-than-average costs and the site selection for renewable hydrogen production becomes critically important. A. Ochoa Bique et al. [44] show how mathematical modeling and mixed integer linear optimization allow to identify optimal decentralized green hydrogen production fields, infrastructure pathways and supply chain networks. In the present study, the green hydrogen is outsourced to FC through R2H2 fields is not counted into the fraction of energy generated with RES. The RES factor contemplates only the portion of energy generated by the RES locally integrated in the CHP-MMG. Hence, it is inferred that the optimal CHP-MMG's configurations are those that store electric energy or convert in thermal energy or exchange it with other grids rather than converting electricity into the hydrogen.

### 5.1. The On-Grid Scenario with Green Hydrogen

This first scenario considers two CHP-MMG connected to the main grid. Each MG can embed a PEMFC-CHP powered with hydrogen generated on site and green hydrogen supplied by R2H2. The optimization is executed using a population of 12 individuals evolving over 220 generations. The datasets are sliced to select one day each month; hence SLSQP executes only 288 loops (LR mode). The remaining generations are computed in HR mode; it means that SLSQP executes 8760 loops. Before switching in HR, the algorithm recalculates the fitness values of the latest j-population to check the consistency of the best-so-far solution. It is remarkable to note that LR computes in 3 min a generation instead of 12 min/generation in HR.

Table 1 shows the geo-locations and the costs of hydrogen used for the scenarios. In Table 2 are listed the price's structures of products (i.e., heat, electricity, and water generated by FC), which are sold to the nearby MG and of the delivered energy demand services to the main grids (GRID). The prices shown are referred to Eurostat 2021 [45].

**Table 1.** Cost of fuel and geo-locations for the simulated scenario.

| Location | Latitude | Longitude | Cost of Hydrogen | |
| --- | --- | --- | --- | --- |
| | | | Actual | Future |
| Bremen | 53.07 | 8.80 | 13 €/kg | 3 €/kg |
| Catania | 37.50 | 15.08 | 13 €/kg | 3 €/kg |

**Table 2.** Price structure of the services offered with CHP-MMG.

| Service/Product | On-Grid | Off-Grid |
| --- | --- | --- |
| Electric energy to user | 38 c€/kwh | 45 c€/kwh |
| Thermal energy to user | 22 c€/kwh | 22 c€/kwh |
| Sellback of electric energy among MG | 5–28 c€/kwh | 23 c€/kwh |
| Sellback of thermal energy among MG | 5–18 c€/kwh | 12 c€/kwh |
| Sellback of electric energy to main grid | 11–18 c€/kwh | |
| Demand response to main grid | −5–28 c€/kwh | |
| Water generated by FC | 15 c€/l | 15 c€/l |

Based on these assumptions, the best solutions (optimal design and sizing) of the best query obtained in the southern (Catania-Italy) and northern (Bremen-Germany) sites, are computed. Table 3 shows the results for the scenarios in each location with the different cost structure of hydrogen. It can be observed, that in Southern Europe, the increase of the hydrogen costs does not affect the optimal mix of RES: the role of PV, WT, ST remains marginal in all cases. Here, a cost of hydrogen 1–3 €/kg, boosts FC to lead the production of electricity and heat. On the contrary, with the actual cost of hydrogen (i.e., 13 €/kg), the optimal supplier of energy remains the GRID with tariff rates that range between 0.24 and 0.36 €/kWh.

The cost of hydrogen heavily impacts the financial performances (IRR) of the optimal CHP-MMG. IRR almost halves when the hydrogen costs doubles. This is mainly due to substantial differences of CAPEX in RES, but similar revenues streams are present in the two scenarios.

In Northern Europe, a high supply cost of hydrogen causes a consistent increment of the PV and WT sizes while FCs are still replaced by main grid. The RES factor, a ratio that correlate the energy generated by RES installed into the MGs with the energy to sell, from 3% jumps up to 100%. The IRR drops by almost 50% if the hydrogen costs raise from 1 €/kg to 3 €/kg. The behavior is justified by a similar structure of net cashflows, yielding no substantial change in IRR.

Table 3. DER size, key financial ratios in different scenarios (25 years project lifetime, 5% discount rate).

| DER | 1.00 €/kgH2 | | 3.00 €/kgH2 | | | | 13.00 €/kg €/kgH2 | | | |
|---|---|---|---|---|---|---|---|---|---|---|
| | Bremen (GE) | | Bremen (GE) | | Catania (IT) | | Bremen (GE) | | Catania (IT) | |
| | MG_A | MG_B | MG_A | MG_B | MG_A | MG_B | MG_A | MG_B | MG_A | MG_B |
| PV | 5 kW | 5 kW | 5 kW | 8 kW | 5 kW | 5 kW | 29 kW | 141 kW | 9 kW | 5 kW |
| WT | 3 kW | 3 kW | 3 kW | 3 kW | 3 kW | 3 kW | 140 kW | 131 kW | 3 kW | 3 kW |
| ESS | - | - | 100 kWh | 14 kWh | - | - | - | - | - | - |
| FC | 12 kW | 18 kW | 41 kW | 40 kW | 13 kW | 10 kW | - | 2 kW | - | - |
| EC | - | - | - | - | 3 kW | - | - | - | - | - |
| OG | - | - | - | - | - | - | - | - | - | - |
| GRID | - | - | - | - | - | - | 100 kW | 100 kW | 22 kW | 100 kW |
| ITCEL | 100 kW | 100 kW | 6 kW | 6 kW | 3 kW | 3 kW | 3 kW | 3 kW | 3 kW | 3 kW |
| ST | - | 5 kW | 5 kW | 5 kW | - | 5 kW | 5 kW | - | - | 5 kW |
| EBOY | - | - | - | 157 kW | 23 kW | 70 kW | 26 kW | 126 kW | 121 kW | - |
| HP | 300 kW | 300 kW | - | 300 kW | 81 kW | - | 281 kW | 281 kW | 242 kW | - |
| STHP | - | - | 300 kW | - | - | 98 kW | - | - | - | 243 kW |
| ITCTH | 5 kW | 5 kW | 5 kW | 5 kW | 5 kW | 5 kW | 5 kW | 5 kW | 5 kW | 5 kW |
| LCOE | 0.15 | 0.161 | 0.155 | 0.223 | 0.136 | 0.116 | 0.114 | 0.165 | 0.164 | 0.172 |
| IRR | 32.7% | 31.7% | 19.7% | 12.6% | 51.1% | 55.4% | 19.9% | 12.4% | 27.4% | 28.3% |
| NPV | 542.586 | 596.401 | 461.218 | 240.464 | 622.123 | 841.975 | 849.888 | 434.543 | 434.533 | 401.888 |
| RES factor | 2.9% | 2.9% | 3.4% | 2.9% | 2.4% | 3.3% | 100% | 100% | 3.2% | 3.3% |

It can be concluded that cost of hydrogen and availability of local renewable resources both deeply influence the optimal MG configuration and IRR. Contrarily, LCOE assumes a similar value in each location and scenarios. This result demonstrates that in different EU locations it is always possible to find an optimal mix of DER that keep the cost of energy at the same level. Hence, these simulations proof the versatility of CHP-MMG technology which can adapt to different climate conditions of South and North Europe. However, this flexibility affects the return on investment: the optimal location for CHP-MMG is Southern Europe.

While the precedent optimizations refer to fixed geo-locations (i.e., Bremen, Catania), a further study was conducted by setting the boundaries of geo-locations in Northern Europe centered in Bremen +/− 7 decimal degree both for latitude and longitude and the same in Catania for South Europe. The scope was to search the best siting and design of CHP-MMG in two regions of Europe with different climate conditions.

These further optimizations were performed with AIE, the same inputs, except for fuel cells for which a more realistic operating condition was set. The start-up/shut-down procedures have been constrained as follows: at startup, the FCs supply a maximum 10% of nominal power and then they must run for at least 1 h before executing the shut-down. Additionally, an average random grid outage at 10% is assumed. It is noted that these conditions affect the search area in both European regions. The sellback to the main grid accounts for over 80% of the overall revenue streams in both regions. Thus, remuneration of energy demand response services has a relevant role in the IRR. When sellback policies are not in place, IRR is null and thus there is not any payback of the investments.

Within the cluster of best solutions, the AIE selects in Denmark the best siting of Northern Europe for CHP-MMG. This region has favorable wind conditions and thus WT are the best choice for RES coupled with FC and the macro-grid. Thermal energy is generated with heat pumps and exchanged between the MGs. Similarly, in Southern Europe the best configurations comprise a mix of PV with WT associated with FC and the main grid. Here the best siting from Catania is the east part of the Mediterranean Area (i.e., Greece) where the combination of solar radiation, wind speed and cloudiness conditions are more favorable.

These simulations have brought out a scheme of collaboration between the two micro-grids (i.e., 'swarm effect') fostering the overall efficiency, energy resilience to uncertainty,

and optimal financial performance [36]. Interconnections are used to permit crossflows of thermal and electrical energies. Notably, a fraction of the exceeding electrical energy (i.e., not used to feed the loads or charge batteries), flows via the interconnections back and it is converted into thermal energy. Thus, a portion of heat swarms to the other microgrid.

*5.2. The Off-Grid Scenario*

The analysis in this paragraph is extended to is-landed CHP-MMG. Hence, in an off-grid scenario it is assumed that the microgrid does not have access to hydrogen refilling as well to the main grid. The refilling of OG is still considered. To simulate these scenarios, the EA boundaries for the main grid have been set to zero (e.g., microgrids disconnected to the main grid). At the initial start-up of the plant, the thermal storage tanks and hydrogen fuel tank are empty and the initial state of charge of the lithium battery is 30%. Based on these harsh conditions it is intended to simulate the resilience of the CHP-MMG at the beginning of the operations, and the financial sustainability of the entire off-grid configurations in Europe. Price structures are indicated in Table 2.

The off-grid optimization problem has solved by EA in two queries without slicing the datasets (i.e., HR resolution). The first query has processed 120 generations. In the second query, 90 generations were enough as the boundaries has been narrowed to values nearby the best solutions given in the first run. The overall results are presented in Table 4.

**Table 4.** Size of DER, key financial ratios for isolated CHP-MMG without hydrogen refilling (25 years project lifetime, discount rate of 5%, off-grid, no daily refilling of hydrogen).

| DER | Bremen (DE) | | Catania (IT) | |
|---|---|---|---|---|
| | **MG_A** | **MG_B** | **MG_A** | **MG_B** |
| PV | 10 kW | 45 kW | 10 kW | 11 kW |
| WT | 60 kW | 79 kW | 111 kW | 109 kW |
| ESS | 49 kWh | 45 kWh | 41 kWh | 50 kWh |
| FC | 0 kW | 0 kW | 0 kW | 0 kW |
| OG | 0 kW | 0 kW | 0 kW | 0 kW |
| EC | 0 kW | 0 kW | 0 kW | 0 kW |
| GRID | 0 kW | 0 kW | 0 kW | 0 kW |
| ITCEL | 21 kW | 21 kW | 74 kW | 74 kW |
| ST | 59 kW | 114 kW | 109 kW | 300 kW |
| EBOY | 0 kW | 0 kW | 0 kW | 0 kW |
| HP | 0 kW | 0 kW | 0 kW | 0 kW |
| STHP | 272 kW | 300 kW | 259 kW | 262 kW |
| ITCTH | 0 kW | 0 kW | 74 kW | 74 kW |
| LCOE | 0.11 €/kWh | 0.12 €/kWh | 0.16 €/kWh | 0.15 €/kWh |
| IRR | 24.70% | 22.70% | 13.60% | 14.00% |
| NPV | 657.634 € | 756.342 € | 391.721 € | 475.883 € |
| RES factor | 57% | 68% | 81% | 100% |

The AIE has selected a similar mix of DER in both sites. It is worth noting that the range of LCOE in off-grid (i.e., 0.11–0.16 €/kWh) is narrower than to that in on-grid scenarios (0.11–0.22 €/kWh). The optimal configuration combines WT, ESS, the electrical interconnection among MG, direct thermal energy generation with ST and its heat hump (STHP). This MG setting is demonstrated to be an effective solution to overcome the missing refilling of hydrogen and the electricity supplied by the main grid. It is noted that both locations do not include FC and OG. The AIE penalizes the candidate configurations generating on-site hydrogen with ECs. Although in an on-grid scenario both choices for the hydrogen supply are considered, configurations that include hydrogen generation on site are not selected (e.g., the size of EC is zero in MG_A and MG_B). In Northern Europe, TCO is lower than on-grid scenarios where the hydrogen cost is 3–13 €/kWh. The revenues generated in this scenario are slightly lower as the demand response services are not furnished. On the contrary, in Southern Europe, TCO are higher than the correspondent

on-grid cases depicted in Table 3. This is explained by the different size of RES which have a lower specific yield than in Northern Europe. In both locations, the deprivation of income given by energy demand services and sellback to a main grid in on-grid, lowers the financial performances. However, the effect is marginal. The IRR remains attractive, i.e., in South Europe it is higher than 13% and 22% in North Europe. Besides, it is noted that RES factors are not 100%, as it should be expected in off-grid contests, without any external outer energy contribution. Indirect generation of energy by STHP through RES is not considered.

These simulations have demonstrated the resilience of the off-grid CHP-MMG system. The SLSQP algorithm is able to perform a full balance of both thermal and electric energies with the PBS. No thermal and electric energy underflows have been computed during the whole-time horizon (i.e., 8760 h). It can be concluded that with a structure of energy tariffs somewhat higher than on-grid CHP-MMG, islanded systems ensure a reliable, regular supply of energy with highly attractive IRR, and convenient LCOE.

*5.3. The "Fuel Cells Powered by Blue Hydrogen" Scenario*

Another set of simulations have been conducted replacing 'green hydrogen' with 'blue hydrogen' in combination with FC. Although green hydrogen leads to the lowest greenhouse gas (GHG) emissions, networks, infrastructures for storage and transport of hydrogen are not yet established. On the contrary, the infrastructure of natural gas containing methane is available worldwide and hydrogen is already produced industrially from steam methane reforming (SMR). The drawback of SMR is the generation of significant carbon emissions. Thus, this type of hydrogen is known as "grey" hydrogen. In the cleaner version (SMR with CCS) named as "*blue hydrogen*", carbon emissions are captured and stored, or reused.

These processes emit few GHG because the reaction from methane to hydrogen yields only solid carbon and no $CO_2$. S. Timmerberg et al. [46] have assesses the life cycle GHG emissions and the levelized costs for hydrogen provision from three type of methane pyrolysis (plasma, molten metal, and thermal gas).

The results of these configurations have been then compared to electrolysis and SMR with and without $CO_2$ capture and storage (CCS). In methane pyrolysis, GHG are between 43 and 97 g $CO_2$ eq./MJ which is mainly caused by the primary energy source. They have concluded that the lowest emissions are obtained with the combination of plasma-based methane decomposition with renewable electricity. This configuration shows lower GHG emissions compared to the "classical" SMR (99 g $CO_2$-eq./MJ), but similar emissions to the SMR with CCS (46 g $CO_2$-eq./MJ).

Two types of plasma can be applied to methane decomposition: cold and thermal (hot) plasma. Cold plasma processes typically show lower conversion efficiencies compared to hot plasma processes. Hence, a further optimization study based on Southern Europe, have been launched again. The latter work intends to analyze the financial performance of optimal CHP-MMG embedding PEMFC-CHP and on-site hydrogen generation thermal plasma (TP) systems operating in on-grid.

Table 5 shows the comparison between two optimal on-grid CHP-MMG configurations delivered by AIE after 350 generations. The first comprises PEMFC-CHP powered by green-hydrogen (i.e., a hydrogen supply by remote R2H2) and the other powered by blue-hydrogen (i.e., hydrogen supplied by local TP systems). Both are located in northern Europe, near Bremen. In the latter, the carbon black is generating a revenue stream being a valuable by-product produced during methane decomposition.

The outcomes are in agreement with the techno-economic analysis prepared by T. Keipi et al. [47] and by R. Dagle et al. [48], concerning the carbon dioxide-free production of hydrogen using natural gas to generate solid carbon and hydrogen.

**Table 5.** Key financial ratios of the on-grid scenarios in North Europe (Bremen), 25 years project lifetime, green hydrogen cost: 3 €\kg (0.25 €\Nm$^3$), blue hydrogen cost: 0.85 €\N m$^3$, 5% discount rate. Revenues and TCO related to the first year.

| | Blue Hydrogen | | Green Hydrogen | |
| --- | --- | --- | --- | --- |
| | MG_A | MG_B | MG_A | MG_B |
| Energy to loads | 33.815 € | 33.265 € | 33.912 € | 33.880 € |
| Energy services to main grid and microgrid | 111.553 € | 49.802 € | 67.130 € | 64.033 € |
| Water by FC | 21.370 € | -€ | 33.097 € | 31.498 € |
| Carbon by TP | 142.465 € | -€ | - | - |
| *Total revenues* | *309.203 €* | *83.067 €* | *134.139 €* | *129.411 €* |
| Total costs of ownership | 1053.678 € | 320.379 € | 268.224 € | 273.174 € |
| LCOE | 0.28 €/kWh | 0.16 €/kWh | 0.21 €/kWh | 0.21 €/kWh |
| NPV | 478.297 € | 239.528 € | 280.000 € | 297.199 € |
| IRR | 9.80% | 12.20% | 17.90% | 17.90% |

The carbon black that has been produced thus far has been used in tires and other rubber products as reinforcing fill (among others), increasing the abrasion resistance of the product. Market prices for carbon black products range from 0.4 to +2.00 €/kgc and can be higher for graphite, CNT. In this presented CHP-MMG problem, the selling price for carbon products is set at 2.50 €/kg$_C$. It should be noted that the TP-based methane decomposition subsystem, yields additional total costs of ownership. Thus, the overall CAPEX of FC integrated with TP are assumed to be: 3.500 €/kW instead of 2.200 €/kW (fuel cell powered directly with TP).

Assuming the cost of methane to 1.20 €/kg$_{CH4}$ (0.85 €/Nm$^3$), electricity tariff at 0.38 €/kWh, a consumption of methane 223 MJ/kg$_{H2}$, 13.90 kWh$_{el}$/kg$_{H2}$ of electricity [46], TP leads to a net hydrogen cost of 2.59 €/kg$_{H2}$ (i.e., cost of productions deduced from the selling price of carbon). The operational costs are also affected by the unfavorable energy conversion efficiency (from methane to electricity) that drops with CHP-PEMFC power with blue hydrogen from 44% to 21%.

Under this assumption and the remaining input data equivalent to those indicated in Section 5.2, the outcome of the CHP-MMG simulations in grid-on configuration do not show cost benefits versus a fuel cell powered with outsourced green hydrogen at 3.00 €/kg$_{H2}$. Nonetheless, the TP yields 3.00 kg$_C$/kg$_{H2}$ which contributes to the revenue stream. The result shows that the overall financial performances (IRR) although lower than the configuration based on green hydrogen is still very attractive. Thus, TP should be considered as a viable technology until R2H2 will not be massively spread.

## 6. Conclusions

In this work, a multi-layer optimization method combining SLSQP with novel self-adaptive differential evolutions algorithms is proposed. The scope is to simultaneously identify the optimal design, configurations, and siting of CHP-MMG while the power dispatch balance is delivered at each time-step. The multi-objective function is formulated to obtain the highest IRR to investors and the lowest LCOE for users. The candidate configuration generated with evolutionary algorithms feeds the analytical techno-economic model.

Two novel self-adaptive evolutionary algorithms based on different mutant and crossover strategies are proposed. The results show that AIE performs 70% better than original DE. An optimization tool has been used to conduct sensitivity analysis of hydrogen costs in two locations placed in different latitudes. In Southern Europe, the optimal mix of RES is selected in combination with FC if the cost of hydrogen is below 3.00 €/kg$_{H2}$, otherwise the main grid replaces the FC.

The overall analysis demonstrates that the cost of hydrogen and the availability of renewable resources affects the optimal design and sizing of CHP-MMG and IRR. The study of CHP-MMG in off-grid scenarios demonstrates that an optimal configuration with LCOE, similar to the on-grid scenarios, can be obtained. Finally, these optimization-

simulations prove the resilience of the off-grid CHP-MMG systems: the best configurations do not exhibit energy underflows. Further optimizations have been carried out with on-site hydrogen generation by TP methane decomposition. This technology leads to a substantial increment of the revenue stream if carbon black is taken into account.

The optimal on-grid CHP-MMG configuration with TP, delivers an IRR: 17% to 21%; this performance leads to the conclusion that it can be considered as a viable "bridge" hydrogen generation technology.

The whole study brings out the versatility of CHP-MMG technology, an important geopolitical issue nowadays. Different settings of operating conditions, costs of fuels, energy demand price policies, and locations are not obstacles to achieving optimal configurations, sizing, and siting of CHP-MMG delivering high quality energy and attractive financial performance.

**Author Contributions:** Conceptualization, P.F., E.Z. and M.F.; methodology, P.F., E.Z. and M.F.; software, P.F.; validation, S.V. (Stanimir Valtchev), S.V. (Svilen Valtchev) and K.C.; formal analysis, S.V. (Stanimir Valtchev) and S.V. (Svilen Valtchev); investigation, P.F.; resources, P.F.; writing—original draft preparation, P.F.; writing—review and editing, P.F., E.Z., M.F., S.V. (Stanimir Valtchev) and S.V. (Svilen Valtchev); All authors have read and agreed to the published version of the manuscript.

**Funding:** This research received no external funding.

**Conflicts of Interest:** The authors declare no conflict of interest.

## Abbreviations

| | |
|---|---|
| ADE | Self-Adaptive Differential Evolution |
| AIE | Self-Adaptive Artificial Immune Evolutionary Algorithm |
| ATE | Analytical Techno-Economic model |
| BSF | Best-So-Far |
| $C^A$ | Convergence of Antibody |
| CAPEX | Capital Expenditure |
| CCS | Carbon Capture and Storage |
| CHP-MG | Combined Heat and Power Microgrid |
| CHP-MMG | Combined Heat and Power Multi-Microgrid |
| CNT | Carbon Nanotube |
| DE | Differential Evolution |
| DER | Distributed Energy Resource |
| DER-CAM | Distributed Energy Resources Customer Adoption Model |
| DG | Distributed Generation |
| DNA | Deoxyribonucleic Acid |
| DNN | Deep Neural Network |
| $D^R$ | Diversity of Repertoire |
| EA | Evolutionary Algorithm |
| EBOY | Electric Boiler |
| EC | Electrolyzer |
| ECM | Evolutionary Computation Method |
| EMS | Energy Management System |
| EP | Evolutionary Programming |
| ES | Energy Storage |
| ESS | Electric Energy Storage |
| EV | Electric Vehicles |
| FA | Firefly Algorithm |
| FC | Fuel Cell |
| FCEV | Fuel Cell Electric Vehicle |
| GA | Genetic Algorithm |
| GHG | Greenhouse Gas |
| GP | Genetic Programming |
| GRID | Utility (main) grid |

| HGT | Horizontal Gene Transfer |
| HM | Hybrid Mode |
| HP | Heat Pump |
| HSDE | Hybrid Differential Evolution and Harmony Search |
| IRR | Internal Rate of Return |
| ITCEL | Electric Interconnection between microgrids |
| ITCTH | Thermal Interconnection between microgrids |
| LCOE | Levelized Cost of Energy |
| LDR | Load Dissatisfaction Rate |
| LSOE | Levelized Sale of Energy |
| MACO | Multi-layer Ant Colony Optimization |
| MG | Microgrid |
| MILP | Mixed Integer Linear Programming |
| MMG | Multi Microgrid |
| MOP | Multi Objective Optimization Problem |
| NLP | Non-Linear Programming |
| NN | Neural Network |
| NOFE | Number Of Fitness Evaluation |
| NPV | Net Present Value |
| OF | Objective Function |
| OP | Optimization Problem |
| OPEX | Operational Expenditure |
| PBS | Probabilistic Best Solution |
| PEMFC-CHP | Combined Heat and Power Proton Exchange Fuel Cell |
| PSO | Particle Swarm Optimization |
| PV | Photovoltaic Panel |
| R2H2 | renewable-to-hydrogen |
| RES | Renewable energy systems |
| RNA | Ribonucleic Acid |
| SLSQP | Sequential Least Squares Programming |
| SMR | Steam Methane Reforming |
| SSER | Small-Scale Energy Resources |
| ST | Solar Thermal Collector |
| STHP | Solar Thermal Heat Pump |
| TCO | Total Cost of Ownership |
| TP | Thermal Plasma |
| VGT | Vertical Gene Transfer |
| WT | Wind Turbine |
| LR | Low Resolution |
| HR | High Resolution |
| OG | Other Generators |

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
