# Peer review of "Techno-Economic Optimization Study of Interconnected Heat and Power Multi-Microgrids with a Novel Nature-Inspired Evolutionary Method"

_electronics, doi:10.3390/electronics11193147_

Round 1
Reviewer 1 Report
The authors carried out a techno-economic analysis of the CHP-MMG system. For this, they used the nature-inspired two-layer optimization algorithm. Many data used in the article are not given. As such, it is impossible for other researchers to verify this article. This hinders the development of a scientific study. Performing the following corrections will improve the article.
1. Contributions and novelty of the study are not clear.
2. The block diagram of the microgrid should be added to the paper.
3. In the introduction, there is no reference. You can look for the above references;
4. What is multi-microgrid? How many areas or units are included in this multi-microgrid?
5. What is the power of the CHP-MMG? What are the properties of these systems? Specifications must be given for both CHP and MMG.
6. What are the values of ka, kb, and kc weights in Eq. 9? How did you determine these weights? Please verify them.
7. How did you determine the iteration number? Please verify this.
8. How did you determine the data in Table 2?
9. Statistical analysis is required and algorithms should be compared in terms of the result of this analysis.
10. What are the equations for the LCOE, IRR, LSOE etc.? What are the penalties in line 438? Please express them.
11. It is not specified what MGA and MGB are. This needs to be explained.
Author Response
REV_01
- Contributions and novelty of the study are not clear.
A. clarification is added in the “Introduction”.
2. The block diagram of the microgrid should be added to the paper.
A. Diagram added in section: “Results and discussion”.
3. In the introduction, there is no reference. You can look for the above references
Referenced added.
4. What is multi-microgrid? How many areas or units are included in this multi-microgrid?
Improved description of CHP-MMG in sections : “Introduction” and : “Results and discussion”.
5. What is the power of the CHP-MMG? What are the properties of these systems? Specifications must be given for both CHP and MMG.
Specifications of power are given in the tables 3 and 4.
6. What are the values of ka, kb, and kc weights in Eq. 9? How did you determine these weights? Please verify them.
Explanation is given in the section: “Fitness and Function”.
7. How did you determine the iteration number? Please verify this.
Improved descriptions in in the section: “Fitness and Function”.
8. How did you determine the data in Table 2?
The reference of Eurostat 2021 is added to justify the electricity prices.
9. Statistical analysis is required and algorithms should be compared in terms of the result of this analysis.
Figure 2 results from a statistical analysis as described in paragraph “The two-layer optimization method”.
10. What are the equations for the LCOE, IRR, LSOE etc.? What are the penalties in line 438? Please express them.
The ref. Nr. 24,38, 40 provide a full explanations of these terms.
11. It is not specified what MGA and MGB are. This needs to be explained.
Figure 3 is added to explains MGA and MGB.

Reviewer 2 Report
Dear authors,
your manuscript is very good. However, please provide more sentences in Section Introduction to figure out clearly what is key contributions of the study.
And in the Section - Results: Please provide a comparison or evaluation between the results from your study and other studies if possible.
Thanks.
Author Response
REV_02
- our manuscript is very good. However, please provide more sentences in Section Introduction to figure out clearly what is key contributions of the study.
A clarification is added in the “Introduction”.
- And in the Section - Results: Please provide a comparison or evaluation between the results from your study and other studies if possible.

Reviewer 3 Report
- The evolutionary algorithm used in this paper should be clarified in its fine-tuning of the parameters for the proposed AIE. The time taken for convergence is also important in the system's efficacy.
- It is necessary to update the references with a review of more recent publications regarding the evolutionary algorithm and its use in microgrids in the Section 2;
Author Response
REV_03
The evolutionary algorithm used in this paper should be clarified in its fine-tuning of the parameters for the proposed AIE. The time taken for convergence is also important in the system's efficacy.
A short explanation is given at the end of the Section: “Self-adaptive differential evolution algorithm”. However, for a detailed explanation We refers to the reference of the author nr. 24.
It is necessary to update the references with a review of more recent publications regarding the evolutionary algorithm and its use in microgrids in the Section 2.
Six more recent works has been considered and added in the paper in the paragraphs:
2.1. Evolutionary computing for optimal operations
2.2. Evolutionary computing for optimal siting and sizing
The reference list has been updated with these new papers.

Reviewer 4 Report
The manuscript treats techno-economic optimization of the combined cool/heat and electricity multi-microgrid systems. A multi-layer optimization method is proposed combining SLSQP with a novel self-adaptive differential evolution algorithm. The multi-objective function is formulated to obtain the highest IRR for investors and the lowest LCOE for users. Two novel self-adaptive evolutionary algorithms are proposed based on different mutant and crossover strategies. The results show that the self-adaptive Artificial immune evolutionary algorithm (AIE) performs 70% better than the original Differential evolutionary (DE) one.
The manuscript is interesting, and the English is good, but the topic is somehow on the borderline of the topics of the usual papers in the Electronics journal.
Here are some comments and suggestions for the manuscript improvement:
1. In the Abbreviation list, add the list of variables, so that it becomes a Nomenclature list.
2. Introduction section should be extended to include an overview of the papers treating the CHP-MG, and CHP-MMG.
3. Contribution of the manuscript should be clearly stated in the Introduction section.
4. Table 2: The basic prices, shown in the table, represent their level in a certain region (or location). However, they might be different in other regions. Furthermore, we are witnessing high variation in energy prices in Europe and the World. Therefore, some comments should be included, with an explanation of the effect of local prices, price variations, and other inputs to the proposed optimization algorithm.
5. The presented results are obtained for certain microgrid structures (several MGs), DGs generation patterns, and certain demand diagrams. However, they are not shown. Give some explanations and add the data in the text.
6. For the locations for the proposed algorithm testing, Bremen in Germany and Catania in Italy are selected. There are no explanations for why these particular locations are selected and which criteria have been used in the selection process. The reviewer is of opinion that other locations may be more suitable for the testing purpose, the ones that have a wider range of temperature and electricity demand variations.
7. There is no comparison of the obtained results with the ones resulting from the application of other (existing) optimization algorithms. Therefore, the conclusions on the proposed algorithm benefits are not convincing. Add the comparison in the text or give some explanations.
Author Response
REV_04
In the Abbreviation list, add the list of variables, so that it becomes a Nomenclature list.
In order to help the reader, the variables are explained directly in the sections where they are used.
Introduction section should be extended to include an overview of the papers treating the CHP-MG, and CHP-MMG.
References are added in the "Introduction".
Contribution of the manuscript should be clearly stated in the Introduction section.
A clarification is added in the “Introduction”.
Table 2: The basic prices, shown in the table, represent their level in a certain region (or location). However, they might be different in other regions. Furthermore, we are witnessing high variation in energy prices in Europe and the World. Therefore, some comments should be included, with an explanation of the effect of local prices, price variations, and other inputs to the proposed optimization algorithm.
The purpose of the simulations-optimisations is to analyse the impact of different price/cost structures applied on average in the EU with reference to Eurostat 2021.However, as we are seeing in recent months, price structures are changing dramatically. In this paper we mainly intend to investigate the best CHP-MMG configurations based on typical price/cost structures and help the reader (e.g. the policy maker) to understand their impact.
The presented results are obtained for certain microgrid structures (several MGs), DGs generation patterns, and certain demand diagrams. However, they are not shown. Give some explanations and add the data in the text.
A clarification of the CHP-MMG framework and figure is added in section: “Results and discussion”.
For the locations for the proposed algorithm testing, Bremen in Germany and Catania in Italy are selected. There are no explanations for why these particular locations are selected and which criteria have been used in the selection process. The reviewer is of opinion that other locations may be more suitable for the testing purpose, the ones that have a wider range of temperature and electricity demand variations.
A Diagram added in section: “Results and discussion”.
There is no comparison of the obtained results with the ones resulting from the application of other (existing) optimization algorithms. Therefore, the conclusions on the proposed algorithm benefits are not convincing. Add the comparison in the text or give some explanations.
The paragraph: “3.2. Self-adaptive differential evolution algorithm” end with a comparison with Differential Evolution Algorithm.

Round 2
Reviewer 1 Report
Some corrections have been made. But there are a few things missing.
1. There are not enough references in the introduction part. This is a scientific article. That's why the things written should be supported by references.
2. In the previous revision, I asked about the values of the coefficients ka, kb and kc in Equation 7. In the article, it is said that it is obtained experimentally. Still not sure what their value is?
3. There is no explanation in the article about LCOE, IRR, LSOE and penalties. I requested in the previous revision that there should be short explanations. But it was said that references can be looked at. These expressions are not ordinary expressions, they form the basis of the study. Therefore, it should be explained briefly.
Author Response
Dear Reviewer,
First of all, I would like to thank all of your valuable contributions and recommendations.
Please find below the comments to your latest recommendations.
- Q: There are not enough references in the introduction part. This is a scientific article. That's why the things written should be supported by references.
A: The "Introduction" section was documented with 6 references based on the key concepts formulated in the text.
2. Q: In the previous revision, I asked about the values of the coefficients ka, kb and kc in Equation 7. In the article, it is said that it is obtained experimentally. Still not sure what their value is?
A: In chapter 4 the explanation of the three coefficients is expanded with further details to clarify their contribution and how they were derived in this work.
3. Q: There is no explanation in the article about LCOE, IRR, LSOE and penalties. I requested in the previous revision that there should be short explanations. But it was said that references can be looked at. These expressions are not ordinary expressions, they form the basis of the study. Therefore, it should be explained briefly.
A: A short explanation how the three terms (LCOE, IRR, P) are computed in the chapter 4 is provided.
Moreover, in 332-335 and 511-519 a short description of their definition is given. The detailed models of this terms are discussed in the references 24 and 40. The latter are mentioned in the chapter 3 and chapter 4.

Reviewer 4 Report
The manuscript has been improved according to the comments and suggestions that were given by this reviewer. There are no further ones.
Thank you.
Author Response
Dear Reviewer,
I would like to thank all of your valuable contributions and recommendations.
